# Delayed Union and Nonunion: Current Concepts, Prevention, and Correction: A Review

**DOI:** 10.3390/bioengineering11060525

**Published:** 2024-05-22

**Authors:** Kristin M. Bowers, David E. Anderson

**Affiliations:** Large Animal Clinical Sciences, University of Tennessee College of Veterinary Medicine, 2407 River Dr., Knoxville, TN 37996-4550, USA; kmbowerseq@gmail.com

**Keywords:** fracture, delayed union, nonunion, bone mechanobiology, dynamization, surgical fracture fixation

## Abstract

Surgical management of fractures has advanced with the incorporation of advanced technology, surgical techniques, and regenerative therapies, but delayed bone healing remains a clinical challenge and the prevalence of long bone nonunion ranges from 10 to 15% of surgically managed fractures. Delayed bone healing arises from a combination of mechanical, biological, and systemic factors acting on the site of tissue remodeling, and careful consideration of each case’s injury-related, patient-dependent, surgical, and mechanical risk factors is key to successful bone union. In this review, we describe the biology and biomechanics of delayed bone healing, outline the known risk factors for nonunion development, and introduce modern preventative and corrective therapies targeting fracture nonunion.

## 1. Introduction

### 1.1. Delayed Union and Nonunion of Fractures

Surgical management of fractures has advanced with the incorporation of advanced technology, surgical techniques, tissue-engineering, and regenerative therapies, but delayed bone healing remains a clinical challenge [1]. The US Federal Drug Administration (FDA) defines fracture nonunion as the persistence of a radiologically visible fracture line at greater than 9 months post-injury [2]. In clinical practice, fractures may be treated as a nonunion as early as 6 months post-injury, and delayed healing can be detected between 3 and 6 months post-injury [2]. Roughly 10–15% of surgically managed fracture cases will result in nonunion. Specific nonunion rates differ depending on the fracture location and fixation, but, in general, these nonunion rates have remained consistent over the past 40 years despite medical and surgical advances [3,4,5]. In a retrospective analysis of US-managed medical claims between 2005 and 2008, the nonunion of an open tibial fracture was estimated to cost roughly USD 25,500 in direct medical costs and 6–12 months of work loss [6]. Nonunion patients also face risks of opioid overuse, medical-related mental illness, and persistent pain even after the fracture union is achieved [7]. Despite the significant direct and indirect impacts of delayed bone healing, our understanding of fracture nonunion remains incomplete, and both preventative and corrective therapies are under development [8]. This targeted review aims to describe the biology of delayed bone healing, outline the risk factors for nonunion development, and introduce advances in the prevention and corrective surgical and medical techniques targeting fracture nonunion.

### 1.2. The Biology and Mechanics of Bone Healing

Successful bone healing relies on a balance of mechanical, biological, and environmental conditions. Mechanical factors include stress across the fracture gap, interfragmentary motion, the gap size, and interfragmentary strains [9]. Biological factors include site-specific vascularity, soft-tissue damage, the availability of osteoprogenitor cells, and hormone and growth factor concentrations at the site of injury [10]. Environmental conditions are determined by patient-dependent factors or comorbidities that can affect bone healing, such as diabetes mellitus, compartment syndrome, chronic disease, or a history of smoking [11]. Modern fracture fixation and regenerative therapies target the mechanical and biological factors of bone healing to optimize healthy bone formation via primary or secondary bone healing. Primary bone healing is defined as the reunion of fractured cortices without the formation of a callus [12]. Osteoid is laid down on the exposed cortices themselves and, through intracortical remodeling, cutting cones establish new Haversian systems across the original fracture line [13]. Thus, absolute stability between the fracture fragments and substantial interfragmentary compression is necessary to achieve union, and modern compression plate fixation has optimized both conditions to promote primary fracture healing [12]. On the other hand, secondary bone healing is defined as cortical union through the formation of a callus. Instead of direct osteoid production originating from the cortices themselves, secondary bone healing relies on a profound periosteal response, the stabilization of the fracture gap through periosteal callus formed via endochondral ossification, bone healing through both endochondral and intramembranous ossification, and the gradual remodeling of the bridged bone to return to its original morphology [13]. A relationship between interfragmentary strain and the type of healing that a fracture undergoes has been discovered, with strains <2% typically fostering primary bone healing, a 2–10% strain encouraging secondary bone union, and a >10% strain predisposing fracture nonunion [14]. In general, fixed-beam constructs such as locking plates elicit secondary bone healing through stable fixation without the degree of interfragmentary compression that compression plating provides [15]. However, recent research posits that the mechanics of successful bone union are dynamic, with ideal strain, hydrostatic stresses, and interfragmentary motion changing throughout the course of bone healing [10,16,17,18,19]. This may account for similar nonunion rates between compressive and fixed-beam constructs.

Surgical fracture fixation serves several mechanical functions to foster bone healing. Fixations transmit force from one end of the bone to the other, allowing for necessary load-bearing functions without excessive motion at the fracture site [15]. These constructs maintain the mechanical alignment of fractured bones, preserving not only the bone’s original anatomy but also the integrity of associated muscles, tendons, and neurovascular tissues [12]. Current options for surgical fracture fixation include external fixation (i.e., Ilizarov-style fixators, hexapod-style fixators, and other circular, hybrid, or linear fixators), intramedullary nail/rod fixation (i.e., the reamed nailing technique, interlocking nails, and Ender nails), internal plate fixation (i.e., neutralization plating, compression plating, locking plate fixation, and point-contact fixators), screw fixation (i.e., position screws and lag screws, either alone or in combination with other internal fixators), and other internal stabilizers such as pins and wires [20,21,22,23]. As mentioned above, surgical fixation aims to stabilize the fracture zone, preserving an anabolic strain environment and protecting fragile granulation tissue, microvasculature, and new cartilage/bone remodeling at the site of injury [24]. Advances in minimally invasive surgical techniques and periosteal-sparing fracture fixations target the biologic factors of bone healing by preserving necessary periosteal and endosteal vascular supply, minimizing disruption to the fracture hematoma and associated signaling factors and reducing infection risk at the site of injury [25,26]. Research into the systemic, patient-associated factors that influence bone healing such as patient age, sex, concurrent diseases (i.e., diabetes mellitus, cardiovascular disease, etc.), concurrent medications, body mass index (BMI), and lifestyle choices (i.e., smoking) has aided clinical decision making and allowed for the refinement of patient-specific surgical techniques [27]. Delayed bone healing and fracture nonunion can arise from any mechanical, biological, or systemic dysfunction, and careful consideration of each nonunion case’s history, presentation, and risk factors will aid clinicians in determining the proper treatment plan.

### 1.3. Classifications of Fracture Nonunion

Following surgical fracture fixation, bone healing is monitored through serial clinical and radiographic evaluation, and, under normal circumstances, improvement should be apparent within three months [28,29,30]. In general, bone healing can be considered delayed between 3 and 6 months postoperative (case-dependent), and, per FDA guidelines, surgical nonunion can be diagnosed after 9 months postoperative [2]. However, due to the multifactorial nature of bone healing and the individual characteristics of each bone, clinical nonunion may be diagnosed earlier than 9 months [29]. On serial radiographic examinations, these cases exhibit a persistent fracture line without progression toward bone union, and patients with surgical nonunion often exhibit associated pain, loss of function, and reduced mobility [28]. Further radiographic classification of nonunion was first described by Drs. Weber and Cech, who classified three general radiographic morphologies of nonunion now termed atrophic, oligotrophic, and hypertrophic nonunion (Figure 1) [8,31,32]. Atrophic nonunion is characterized by a lack of periosteal callus formation and are hypothesized to be associated with impaired vascularity at the fracture site [4]. Oligotrophic nonunion exhibits mild periosteal callus formation and historically were considered to have preserved vascularity but reduced osteogenesis [8]. Hypertrophic nonunion is characterized by excessive periosteal callus formation, often termed “elephant’s foot” or “horse’s foot”, and these nonunions have been associated with fixation instability [4]. Despite widespread use of the Weber–Cech classification system in clinical practice, the inference of vascular and/or metabolic activity of an individual nonunion based on radiographic appearance should be approached with caution [4]. Histological analyses comparing atrophic, oligotrophic, and hypertrophic nonunion, have documented similarities in vascular density and connective tissue ingrowth at the fracture site, and transcriptomic analyses of various tissue samples from nonunion cases yields similar transcriptomes regardless of the Weber–Cech classification [4,8,33]. As advances continue in patient diagnostics, especially in transcriptomic and genomic profiling, fracture nonunion may be reclassified as either mechanical or biological in origin, aiding in successful, case-specific intervention [4,8,34].

### 1.4. Nonunion Risk Factors

Risk factors for delayed bone healing and eventual nonunion can be injury-related, patient-dependent, surgically related, or mechanical (Figure 2). As with all illnesses and injuries, the systemic health of the individual patient plays a substantial role in the outcome of a medical event. Retrospective patient outcome analyses have reported that fracture nonunion may be associated with pre-existing conditions such as diabetes mellitus, cardiovascular disease, immune-mediated disorders, renal insufficiency, cancer, and other immunosuppressive conditions [4,11,19,27]. Certain concurrent medications including chemotherapeutics, anticoagulants, opioids, non-steroidal anti-inflammatory medications, and anabolic steroids have been associated with delayed or arrested bone healing [4,35,36]. In a meta-analysis of the association between a history of smoking and surgical nonunion, Mahajan et al. confirmed a strong association between smoking and complications in bone healing, utilizing data from 12 studies published between 1999 and 2020 (both prospective and retrospective) in their analysis [37]. Patient age has been associated with nonunion but not in a linear fashion. Several studies have noted a greater incidence of tibial and femoral nonunion in younger patients, who commonly present with high-energy, comminuted, and often open fractures [7,8,11,38]. However, other studies have noted increased incidence of nonunion in middle-aged patients (aged 45–65 years) or in patients of advanced age often presenting with pre-existing conditions [3,30]. Thus, patient age may not represent a risk factor for fracture nonunion in isolation, but age-related comorbidities should be considered.

Increased patient body mass index (BMI) has been identified as a risk factor for delayed bone healing and surgical nonunion of long bone fractures [4,27,39]. In a retrospective analysis of nonunion following lateral locking plate fixation of distal femoral fractures, patients classified as obese (BMI > 30) were at significantly greater risk of undergoing a secondary surgical procedure to address delayed bone healing [39]. Additionally, a link between a patient’s compensable/insurance status and hospital readmission for fracture healing complications was identified in a retrospective analysis of humeral, tibial, and femoral (excluding proximal) fractures registered by the Victoria Orthopedic Trauma Outcomes Registry between 2007 and 2011 [3]. Ekegren et al. noted that patients with compensable healthcare plans were 2.43 times more likely to be treated for fracture nonunion than those using Medicare or another non-compensable plan [3]. Although patients’ insurance statuses do not directly affect bone healing, there is evidence that patient socioeconomic status affects healthcare, including the treatments provided, access to providers, routine follow-up, and patient–provider interactions [40,41]. In a qualitative research study investigating American low-socioeconomic-status patients’ perceptions of hospitalization, discharge, and post-hospital transition, Kangovi et al. noted poor compliance to discharge instructions due to economic constraints, availability of care, the lack of targeted follow-up, and the initial misalignment of patient and care team goals [42]. Thus, patient socioeconomic status can be considered a risk factor for bone healing complications leading to nonunion.

Several fracture characteristics have been identified as risk factors for complications of bone healing leading to surgical nonunion. As noted above, younger patients more often present with high-energy-impact fractures characterized by extensive comminution, marked soft-tissue damage, and open fracture status [43]. Open fractures are at significantly greater risk of infection-related nonunion, particularly in cases with severe vascular damage and/or segmental bone loss [44,45]. Several studies have identified significant associations between the risk of fracture nonunion and increasing severity of comminution, as evaluated using the AO/OTA Fracture and Dislocation Classification system described by Meinberg et al. [7,38,46,47]. Further investigation into the effects of fracture configuration was conducted using the finite element analysis modeling of tibia fractures stabilized using intramedullary nail fixation [11]. When all other mechanical and surgical factors were held constant, wedge and complex (marked comminution) fracture geometries exhibited excessive strain patterns that delayed simulated healing, indicating that interfragmentary mechanics could directly affect bone healing regardless of external mechanical conditions [11]. In addition, both in vitro and in vivo analysis of the fracture gap following intramedullary nail fixation of long bone fractures has shown a direct relationship between an increasing fracture gap and risk of delayed bone healing [11,48,49]. Fracture gap and segmental bone loss form the foundation for determination of critically sized defects in clinical fracture management and in the preclinical modeling of delayed bone healing [50,51,52]. These injury-related risk factors are inherently connected with both surgically related risk factors and fracture mechanics in the postoperative period.

Surgical risk factors for fracture nonunion include unstable fixation and inadequate fracture reduction [11]. Determination of the appropriate implant, proper surgical techniques, and an optimal postoperative rehabilitation plan for each fracture is the foundation of modern fracture research, and this review will present several preventative and corrective surgical techniques targeting nonunion complications [53]. Changes in fixation stability through screw loosening, implant breakage, or implant displacement increase the risk of delayed union and often result in necessary revision surgery [2,54]. Mechanical factors in the postoperative period also influence the quality of bone healing. Claes et al. described correlations between the hydrostatic strain and pressure conditions of a fracture and the types of tissues that will be deposited in the fracture callus [10]. Deviatoric stresses exert directional strain across a fracture gap, stimulating fibrous tissue production, whereas hydrostatic stresses exert pressure at the fracture site, stimulating cartilage formation [10]. Mechanical overload in the form of excessive weight-bearing or localized supra-physiologic strain concentrations on the bone itself will delay healing [55,56]. These effects can be compounded by the type and placement of fracture fixation; for example, when locking plate fixations bridging FEA-modeled tibial fracture gaps were experimentally loaded, the greatest stress concentration occurred at the screws closest to the fracture gap, and in cases of short working lengths, stress dissipation led to mechanical overload at the fracture site [57]. In addition, multi-axial loading can contribute to delayed bone healing depending on the type and stability of fracture fixation; for example, intramedullary nail (IMN) fixation is less stable than other fixation methods under torsional stresses, and torsional instability has been identified as a risk factor for nonunion in IMN cases [7,19]. Recently, a meta-analysis of allograft reconstructions compared patient outcomes after fracture stabilization with either a single bridging plate or with an intramedullary interlocking nail. Thirteen studies representing 431 patients were analyzed. Interestingly, the use of bridging plates had significantly fewer cases of nonunion as compared with use of an intramedullary nail (12% vs. 37%). Other morbidities were similar between the groups. Locking plates have similarities to bridging plates, with the exception of the distribution of mechanical forces relative to the bone versus the plate constructs [58]. The interplay between patient-specific, surgical, mechanical, and injury-related factors will determine the success of fracture healing, and increasing understanding of these risk factors has spurred development of several techniques and devices aimed at reducing nonunion prevalence in the future.

## 2. Preventative Techniques and Advances

### 2.1. Dynamization

Dynamization of fracture fixation is an increase in interfragmentary motion delivered in a controlled manner to foster secondary bone healing [59]. It is based on the theory that, while marked fixation stiffness is beneficial to stabilize the fracture hematoma and develop granulation tissue, a degree of controlled motion is necessary to promote callus maturation and endochondral ossification [59,60]. As the use of fixed-angle constructs has increased, an association between locking plate fixations and surgical nonunion has been identified [39,61,62]. In an analysis of the effects of locking plate construct stiffness on fracture healing, Bottlang et al. provided both in vitro mechanical and in vivo clinical evidence that locked-plate constructs may be too stiff to consistently promote fracture healing regardless of the plate length or fixation working length [63]. Thus, dynamization aims to manipulate the biomechanical conditions of the fracture site, often in accordance with the stage of fracture healing [59]. Two simple examples of fracture dynamization are an increase in the fixation working length and the use of semi-rigid locking screws during the locking plate fixation of experimental 10 mm distal femoral osteotomies [64]. In this model, both the working length (regardless of rigid or semi-rigid locking screws) and screw type contributed to axial interfragmentary motion, and the substitution of semi-rigid locking screws during the locking plate fixation of fractures provided auxiliary, controlled interfragmentary motion to stimulate healing [64]. Another example of fracture dynamization is the use of far-cortical locking screws (Figure 3), introducing a degree of micromotion at the near-cortex while maintaining the stability of a fixed-angle construct [61,65]. Plate dynamization has been achieved through elastically suspended locking holes (termed active locking plates), which allow screws to independently slide within a small pocket in the plate, providing axial motion without significantly depreciating the construct strength (Figure 3). Beltran et al. discussed the advantages of stress modulation with fracture fixation implants as a way of decreasing the complications associated with over-stiff implants. Increased flexibility in fracture repair is expected to decrease implant-related morbidities and improve fracture healing. When using bridging plates, strategies such as far-cortical locking and near-cortical over-drilling may be employed to achieve this effect. These will be discussed in detail below [66]. Dynamization using active locking plates and semi-rigid locking screws have reported success in animal models of delayed bone healing, but, currently, only the far-cortical locking screw technique has been evaluated and successful in human clinical trials [61,67,68]. Related clinical trials are ongoing, and the further development and refinement of plate dynamization is expected with the current advances in additive manufacturing [60].

Recent research in bone mechanobiology has indicated that the ideal fracture mechanical environment changes over time, moving from greater stiffness and less strain to more interfragmentary motion as the fracture callus matures [69]. Thus, temporal dynamization aims to induce appropriate interfragmentary motion in accordance with the progression of fracture healing, moving from rigid fixation to more flexible over time [24,67]. The following three general methods of temporal dynamization are either in use or being developed at this time: staged surgical interventions, modifiable external fixation, and degradable biomaterials incorporated in surgical fixations [59,60]. Staged surgical interventions are the most simplistic form of temporal dynamization, in which an initially rigid implant is surgically manipulated after initial callus formation to induce interfragmentary motion [60]. Examples of this include the removal of a distal locking screw from an IMN fixation or removing paired screws adjacent to the fracture in a locking plate fixation to increase the working length [70]. Modifiable external fixation originated as a form of staged surgical intervention, in which an external fixation could be converted from a static to a dynamic state with the removal of stabilizing pins, screws, and/or connecting bars [71,72]. However, external fixators provide an opportunity for more active control of the interfragmentary motion, and additional forms of dynamization, including the use of sliding elements, dynamized pins and screws, and other external controls, have been reported in research assessing the optimal timings and degrees of dynamization during fracture healing [60,73,74].

The temporal dynamization of internal fixators is also under development, and, currently, the most promising modality is the use of biodegradable polymers in plate fixation. Several methods have been described, including the use of a threaded degradable polymer between the locking screws and plate that degrades to induce motion around the screw head, and the use of degradable coatings or composites in the locking plate itself that also induces motion around the screw heads with degradation [59,75]. The Variable Fixation Locking Screw (VFLS, Biomech Innovations AG, Aarbergstrasse, Nidau, Schweiz) combines the concepts of far-cortical locking screws with temporal dynamization [76]. It utilizes a biodegradable poly-(lactide-co-glycolide) (PLGA) sleeve positioned below the screw head to sit within the near-cortex and induce near-cortical micromotion as it degrades (Figure 3) [77]. The initial results of temporal dynamization have been promising in animal models, but further clinical use is limited by both the questions and the risks surrounding this technology [69,71,77]. First, the introduction of interfragmentary motion carries an inherent risk of delayed healing due to excessive motion or interfragmentary strain during early callus formation. Repeated disruption of the fibrous and cartilaginous portions of the fracture callus will impair intramembranous ossification, and the primary goal of fracture fixation is to stabilize the fracture site to promote bone healing [24]. Thus, both the degree and timing of induced interfragmentary motion is undetermined at this time [78]. Early dynamization in a rat femoral ostectomy model resulted in no difference in osteotomy volume, tissue mineral density, or bone mineral density between temporally dynamized and flexible control specimens, suggesting no benefit of early dynamization in this model [72]. With the finite element analysis of temporally dynamized ostectomies, Fu et al. also noted delayed bone healing with early dynamization (within 1 week of experimental fixation), but this effect could be reversed if the magnitude of early dynamization (the amount of motion) was decreased, resulting in the enhanced bone formation and biomechanical strength of the new bone [79]. Further biomechanical and preclinical research aimed at identifying the proper timing and degree of dynamization is ongoing, and clinical research utilizing both static and temporal dynamization techniques are in progress.

### 2.2. Reverse Dynamization

As the name suggests, reverse dynamization is a theory that, in contrast to dynamization, bone healing could be enhanced through the early introduction of interfragmentary motion followed by conversion to more rigid fixation [60]. This stems from the idea that the callus size is directly affected by the degree of interfragmentary motion during the inflammatory phase of healing, and, if a fracture is stabilized early by a larger callus, it will remodel more effectively and quickly [80]. Glatt et al. reported accelerated and improved bone healing with reverse dynamization in a rat femoral ostectomy model comparing static low-stiffness external fixators to reverse-dynamized fixations in the presence of implanted bone morphogenic protein-2 (BMP-2) [81]. However, in some cases, reverse dynamization resulted in the persistence of a cartilaginous union in rat femoral ostectomies treated with BMP-2, indicating that, similar to dynamization, the timing and magnitude of interfragmentary motion will affect the quality of bone healing [80]. While additional reports of improved bone healing using the reverse dynamization of external fixators have been published, further research into internal fixation options for reverse dynamization, as well as the biologic and mechanical processes underlying this methodology, is ongoing [60,82,83]. At this time, publications relating to the in vivo use of reverse dynamization have been limited to preclinical segmental defect models, but clinical translation of these concepts to varying fracture configurations and fixations is a promising next step [60,82,84].

### 2.3. Genomics, Transcriptomics, and Proteomics

Extensive research has described the molecular nuances of fracture healing, but recent advances in genomic, transcriptomic, and proteomic analyses has expanded our understanding of the robust interplay of genetic up- and downregulation at the site of fracture remodeling [85,86,87,88,89]. For example, single-cell transcriptomic analysis has characterized the differences in regulatory gene expression between chondroclasts and osteoclasts at the site of a fracture, and it identified several highly upregulated chondroclast genes (PSMD2, ATP5B, MT-CO1, and GLUD1) that may maintain cellular metabolic activity, thus driving endochondral ossification within a callus [86]. While the extensive analysis of the genomic control of bone healing is outside the scope of this review, recent research targeting genetic or transcriptional markers of fracture nonunion provide promising data for future prognostic biomarkers or perioperative treatment targets [8,87,90,91]. For example, Hadjiargyrou et al. compared the microRNAome of intact bone to fracture callus and nonunion tissue samples and identified both miRNAs unique to each tissue sample, and miRNAs that were co-expressed in all circumstances [8]. These data allowed them to identify target genes differentially expressed between callus and nonunion samples, and these genes could serve as future targets for osteoregenerative therapeutics [8]. Another study utilized genomic analysis to try to understand the link between diabetes mellitus (Type II) and delayed bone healing; Liu et al. identified several modules and hub genes of interest, particularly ANXA3, that may serve as prognostic biomarkers for fracture nonunion in patients with diabetes [91]. As our understanding of the genetic controls of bone healing and transcriptional alterations associated with nonunion advances, we predict that more nonunion-risk biomarkers and preventative therapeutics will be developed for clinical use.

## 3. Corrective Techniques and Advances

### 3.1. Exchange Nailing

Exchange nailing has been an effective treatment for surgical nonunion following IMN fixation, with success rates ranging from 78% to 86% in femoral shaft nonunion [4,92]. An exchange nailing procedure consists of the removal of the indwelling nail, longitudinal drilling of the intramedullary canal, and IMN replacement with a larger-diameter nail [93]. The thicker nail improves the biomechanical stability of fixation and the reaming procedure stimulates both inflammatory and regenerative cell migration to the site of injury through vascular and tissue damage [4]. Exchange nailing is a relatively simple surgical procedure that does not expose the fracture site, typically elicits minimal blood loss, and the hospital stays for an exchange nail procedure are relatively short [43]. However, careful case selection is key to success, as exchange nailing procedures alone have poor union rates in cases of septic nonunion or excessive fracture gaps (>5 mm) [43,93].

Intramedullary reaming represents a key step in exchange nailing, and it is often incorporated into original IMN surgical fixation as well [94]. However, the balance between the benefits and risks incurred by intramedullary reaming remains debated today [94]. Intramedullary reaming disrupts the endosteal blood supply to the bone in question through direct vascular damage and penetration of the central venous sinus within the medullary cavity [95]. Vascular injury directly stimulates tissue regeneration through the chemotaxis of inflammatory cells and mediators, but, for a period of time, vascular supply to the bone is impaired, driving rapid reorientation of blood flow to the periosteum [95]. Concerns regarding both compressive and thermal damage during reaming have been voiced, and the risk of the systemic embolization of bone marrow via venous outflow during reaming has been reported [95,96,97]. However, the value of intramedullary reaming to optimize bone–nail congruence and maximize fixation stability cannot be understated [95,96]. At this time, consensus regarding the use of intramedullary reaming in initial IMN fixation has not been reached, and the majority of clinical studies display marked statistical fragility, with merely one or two event reversals necessary to reverse a binary conclusion [94]. On the other hand, intramedullary reaming during exchange nailing has been refined and augmented over time, serving as a valuable debridement tool for septic nonunion and a delivery method for intramedullary lavage and antibiotic therapy [98]. Clinical research utilizing exchange nailing in conjunction with biologic and regenerative therapies is ongoing and offers promising future paths for the treatment of both aseptic and septic nonunion [93,98,99].

### 3.2. Nail Dynamization

Nail dynamization involves the removal of proximally or distally locked screws from an intramedullary nail fixation to introduce a degree of interfragmentary motion at the fracture site [100]. As discussed above, this procedure can be staged as a preventative measure or enacted as a treatment for diagnosed delayed healing [4]. It represents a relatively simple and inexpensive treatment option for delayed bone healing, and, historically, success rates following nail dynamization in tibial and femoral shaft nonunion (between 2011 and 2014) were reported between 55 and 65% [101]. In clinical practice, nail dynamization may be elected as an early intervention for cases with delayed bone healing, and if bone union is not achieved, more invasive procedures such as exchange nailing may be warranted [102]. As with prophylactic dynamization, nail dynamization procedures carry risks of excessive motion at the fracture site, resulting in fixation failure, fracture shortening, and/or refracture [4,101]. Modern advances in nail dynamization have centered around supplementary therapies such as shock wave to improve bone healing following the dynamization procedure, but recent clinical studies have confirmed the value of nail dynamization as an early intervention for delayed bone healing, reporting improved success rates of 75–85% in tibial and femoral fracture cases [102,103,104].

### 3.3. Augmentation Plating

Augmentation plating describes the addition of an orthopedic plate to a pre-existing fracture fixation (most often IMN) to enhance stability and interfragmentary compression [103]. Augmentation plating can be used alone or in combination with exchange nailing, bone grafts, and/or other biologic therapeutics [4,105,106]. Since it is so versatile, success rates for union following augmentation plating vary based on the chosen methodology, but in a retrospective review of patients diagnosed with femoral shaft nonunion following IMN fixation and treated with plate augmentation (with both dynamic compression and locking plates utilized), Uliana et al. reported an 86% union rate and excellent patient pain and mobility scores over two years of follow-up [107]. Similar successes in treating femoral nonunion with either locking or compressive augmentation plates has been reported in other prospective and retrospective analyses, and this modality has become increasingly popular in the management of femoral nonunion [108,109,110].

### 3.4. Strain-Reduction Screws

Similar to augmentation plating, the use of strain-reduction screws in the treatment of surgical nonunion avoids the removal of the initial fixation [109]. It involves the placement of standard cortical screws across a nonunion site to induce interfragmentary compression and reduce local strains [111]. Currently, clinical reports utilizing strain-reduction screws are few, with use limited almost exclusively to lower limb aseptic nonunion [111,112,113]. However, clinical success has been reported following strain-reduction screw placement across hypertrophic humeral nonunion, in which the initial fixation exhibited no loss of stability [111]. Further large-scale clinical research is necessary to confirm the clinical efficacy of strain-reduction screws and establish their indication for use, but this technique offers a promising minimally invasive option for surgical nonunion treatment.

### 3.5. External Fixation

External fixators are highly customizable and stable fixations that are widely used to treat surgical nonunion, particularly in cases of septic nonunion [114]. They operate percutaneously, and surgical stabilization, using an external fixator, results in minimal soft-tissue and periosteal trauma [114]. External fixator designs allow for a wide range of mechanical stabilization, from linear, unilateral stabilization to circular, multi-axial support, and external fixators can be manipulated to act as fixed-beam, compressive, or dynamized constructs [4,73,74]. External fixators are particularly suited for cases involving severe soft-tissue damage, excessive bone loss, and/or infection [115]. Their versatility allows for staged soft-tissue and bone reconstruction procedures, and percutaneous fixation minimizes the risk of implant-related bacterial dissemination or biofilm formation [114,116]. Recent studies have highlighted the utility of the Ilizarov technique of external fixation, which utilizes the theory of distraction osteogenesis and applies controlled tensile stress to a fracture or nonunion site to encourage tissue regeneration [117]. Ilizarov external fixators stabilize and promote the healing of large bone defects formed by high-impact trauma, extensive resection of infected tissue, or osteotomy to address fracture shortening [117,118,119]. Although limited by surgical complexity, long treatment times, and general inconvenience to the patient, Ilizarov external fixators have reported success rates ranging from 76% to 100% in clinical case reports of long bone septic nonunion, both as solo fixations and in conjunction with other nonunion treatments such as exchange nailing or bone grafting [117].

### 3.6. Bone Grafts

Bone grafts are widely used in the treatment of surgical nonunion, with treatments ranging from cancellous autografts for osteoinduction and the revascularization of an atrophic nonunion to large cortical allographs for osteoconduction and the bridging of critically sized defects [106,120]. While extensive discussion of bone grafting is outside of the scope for this review, we aim to highlight current advances and discussions related to bone graft treatments of surgical nonunions. To start, the topic of graft vascularization stems from a long-standing limitation of bone grafts, particularly allografts, in that the grafted bone lacks blood supply in the implantation site [120,121]. The majority of grafted bone will necrose, serving more signaling and scaffold functions than direct incorporation, and impaired vascularity or diminished osteoprogenitor cell populations at the implant site predispose the patient to graft failure or rejection [120,121]. The use of vascularized autografts involves careful preservation of the graft’s nutrient, metaphyseal, or other perforating vessel for anastomosis at the implant site. If the blood supply is preserved, the graft may incorporate by either primary or secondary bone healing, and vascularized bone grafts deliver donor osteocytes and osteoprogenitor cells more effectively than non-vascularized autografts [120,122]. However, the balance between the osteoinductive benefits of vascular grafting versus the technical difficulty and intensity of the procedure has been debated, spurring several recent, large-scale meta-analyses and systematic reviews of vascularized bone grafts to treat scaphoid nonunion [123,124,125]. These reviews documented favorable results of vascularized bone grafts (more rapid bone union) in cases with poor prognostic indicators, such as systemic disease or avascular necrosis of the scaphoid [124,125]. However, functional results were not as uniformly favorable, and questions remain as to the functional advantage of vascularized bone grafts over non-vascularized grafts in scaphoid nonunion [123].

Currently, the use of vascularized bone grafts is limited by donor site availability and graft viability during auto- or allotransplantation [120]. However, recently, Visser et al. described a technique for the revascularization of cryopreserved bone allografts involving the surgical placement of the host cranial tibial arteriovenous bundle within the intramedullary canal of the donor tissue (porcine tibia model) [121]. The goal was to supply and utilize the vascular channels already present within the graft tissue to encourage direct revascularization and incorporation of the graft. Seven of eight tibial defects treated with revascularized cryopreserved grafts healed within the 20-week experimental period, as compared to four of eight control defects treated with traditional cryopreserved grafts [121]. However, no differences in the bone mineral density or biomechanical characteristics were noted between the experimental and control bones, suggesting that, while revascularization does not seem to impair bone healing, it cannot be proven to enhance bone density or strength from this study alone [121]. This theory of graft revascularization is a promising subject for future research and refinement, and may address limitations such as donor site morbidity, graft availability, and graft rejection.

### 3.7. Tissue Engineering

As with bone grafts, extensive discussion of the numerous bioactive biomaterials, tissue regenerative therapeutics, and cell therapies developed to augment bone healing is outside of the scope of this review. However, we would like to highlight some of the modern focuses of regenerative research targeting surgical nonunion. The field of regenerative bone biomaterials has undergone expansive growth in the last two decades following the approval of calcium phosphate cement for clinical use [126,127]. Ceramic-, polymeric-, and composite-based bone biomaterials have been developed to serve numerous functions, including the osteoconductive filling of bone gaps, drug/hormone delivery, cellular activity direction, and extracellular matrix templating [126]. Bone biomaterials have been used to deliver concentrated and controlled doses of known anabolic (i.e., parathyroid hormone analogues and bone morphogenic protein analogues) and anti-catabolic (i.e., bisphosphonates, calcitonin, denosumab, and estrogen) therapeutics to the sites of nonunion, and modern considerations for bone biomaterial fabrication are presented in Figure 4 [126,128,129,130]. Recent research regarding the use of teriparatide, a synthetic recombinant parathyroid hormone, delivered via a degradable biomaterial for the treatment of fracture nonunion has shown favorable results in both preclinical and clinical trials [129,131]. Further development, refinement, and clinical evaluation of bone biomaterials for use in surgical nonunion cases is ongoing, and more commercial approval and widespread use is expected in the near future [127].

Stem cell therapy (alternatively termed stromal cell therapy) is another modality used in combination with other stabilizing and/or reconstructive procedures to address surgical nonunion [132,133]. Similar to bone biomaterials, there are numerous types of cell therapies currently under research, varying in origin, cellular processing, preconditioning, delivery method, and supplementation (i.e., co-delivery with growth factors or chemokines) [132,134]. Despite the heterogeneity of treatment options, promising clinical results of stem cell therapy as a nonunion treatment have been reported [135]. Following implantation of autologous expanded human mesenchymal stromal cells (MSCs) from bone marrow using a bioceramic delivery device to treat long bone nonunion, femoral, humeral, and tibial nonunion exhibited rapid, effective bone union following implantation, with a 92.8% success rate at 12 months postoperative [135]. In another case series evaluating a combination therapy using autologous stomal cells loaded on collagen microspheres and delivered via platelet-rich plasma clots, rapid and effective bone union was seen following both tibial and femoral applications of the novel therapeutic [136]. However, the clinical efficacy of stem cell therapy in nonunion cases remains questionable, with no consensus as to the source, dosage, processing, or delivery of cells [132,133,134]. Current research exploring the paracrine effects of cell therapy and evaluating the efficacy of MSC exosome implantation aims to further refine the therapeutic use, and further development of cell therapy for clinical use is ongoing [132,137].

### 3.8. Rehabilitation of Patients Suffering Delayed Healing and Nonunion Fracture

Rehabilitation programs are an essential process to speed up and improve patient recovery from fractures. A detailed review of rehabilitation practices is beyond the scope of this review article; however, the value of evidence-based rehabilitation programs cannot be over-emphasized [138,139]. The growing body of evidence supports that patients are more likely to recover fully or to a greater extent when rehabilitation programs are incorporated into their postoperative recovery period [140,141,142]. Effective rehabilitation programs require collaboration among the patient, rehabilitation team, and physicians overseeing the patient’s care. Blackburn and Yeowell conducted a thematic literature review to explore patients’ perceptions of their rehabilitation after having surgery for a hip fracture [143]. The authors concluded that the perspective and experience of the patient are essential elements to be considered when adapting individualized rehabilitation programs for specific conditions.

## 4. Conclusions

Delayed healing and nonunion remain clinical challenges in the surgical management of fractures, with nonunion prevalence ranging from 10 to 15% of fracture cases and the costs associated with care averaging roughly USD 25,500. Delayed bone healing arises from a combination of mechanical, biological, and systemic factors acting on the site of tissue remodeling, and careful consideration of each case’s injury-related, patient-dependent, surgical, and mechanical risk factors is key to successful bone union. Recent advances in fixation dynamization, both static and temporal, and reverse dynamization have the potential to reduce nonunion prevalence upon future refinement and implementation, and discoveries in the genomic, transcriptomic, and proteomic profiles of bone nonunion may direct the development of risk biomarkers and targeted therapeutics for delayed healing. Modern treatments for nonunion include exchange nailing, nail dynamization, augmentation plating, strain-reduction screw placement, bone grafting, and external fixation. These modalities can be augmented by tissue-engineered therapeutics, including bioactive biomaterials, stem cell therapy, anabolic therapies, and anti-catabolic therapies. Thus, modern advances in the mechanics of fracture management combined with tissue engineering and biomaterial innovations provide promising options for both the prevention and treatment of surgical nonunion.

## Figures and Tables

**Figure 1 bioengineering-11-00525-f001:**
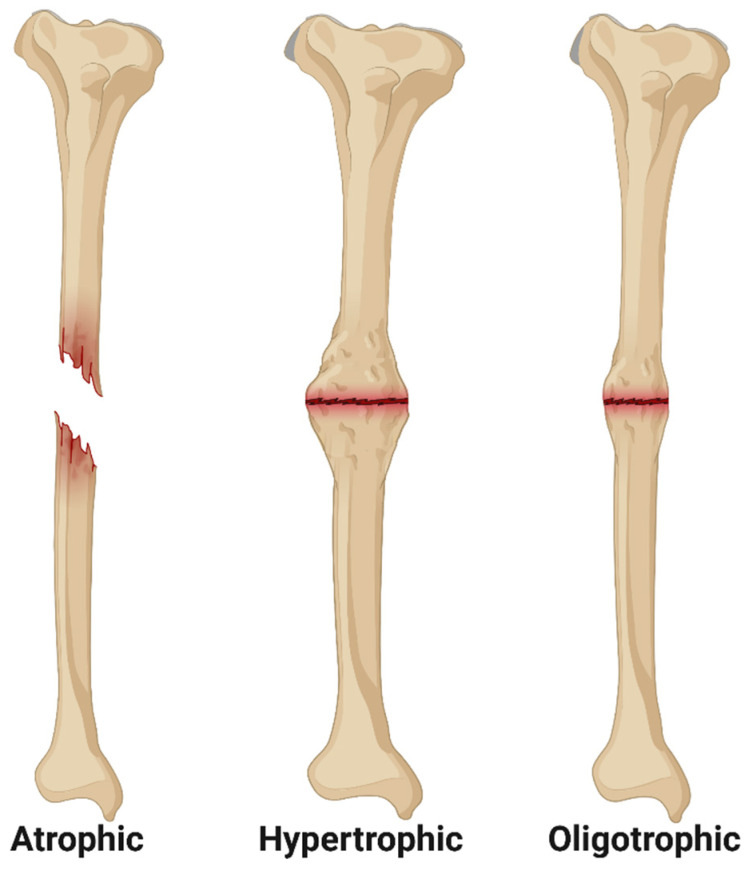
Illustration of the Weber–Cech Classifications of Nonunion. Atrophic nonunion is characterized by no callus formation and often associated with impaired vascularity. Hypertrophic nonunion is characterized by excessive periosteal callus formation without union of the fracture fragments. Oligotrophic nonunion is characterized by mild periosteal callus formation without union of the fracture fragments. Image created using Biorender.com (accessed on 1 March 2023).

**Figure 2 bioengineering-11-00525-f002:**
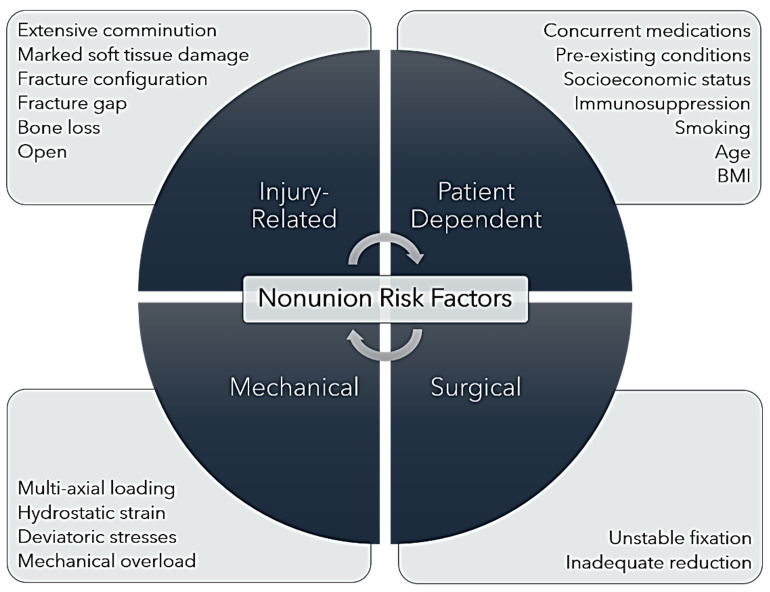
Nonunion Risk Factors. Illustration of the injury-related, patient-dependent, mechanical, and surgical risk factors that contribute to fracture nonunion.

**Figure 3 bioengineering-11-00525-f003:**
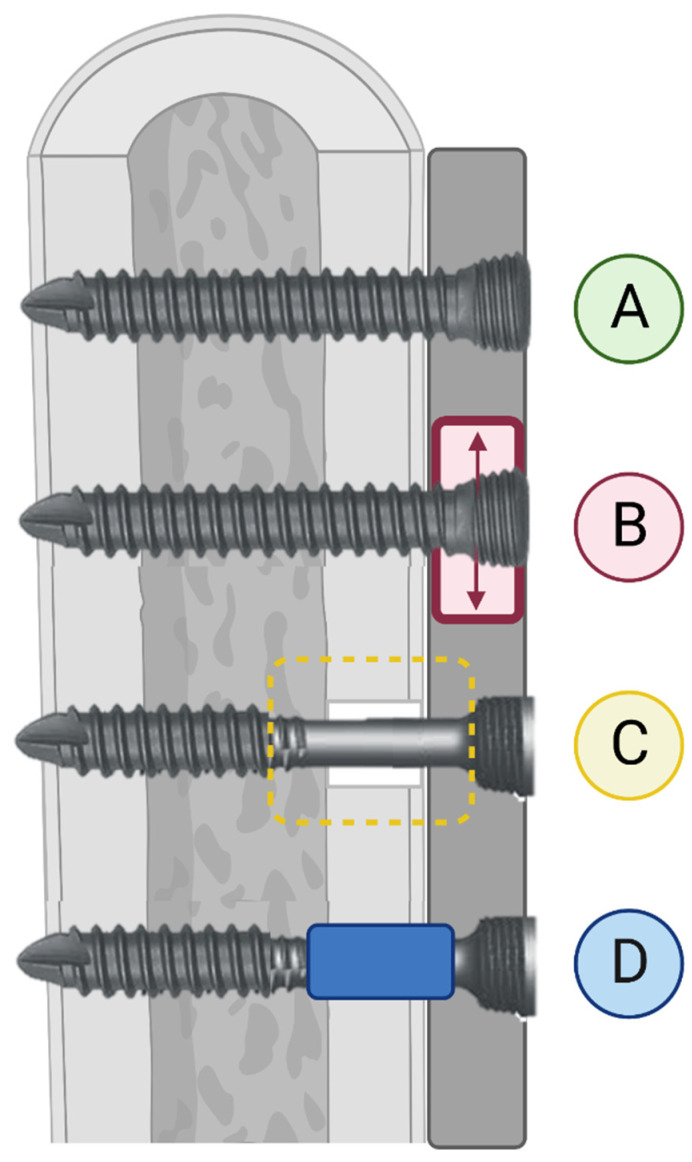
Illustration of Three Methods of Fixation Dynamization. Three methods of dynamized locking plate fixation are depicted as compared to a non-dynamized locked screw. (**A**) Locked screw, non-dynamized. (**B**) Schematic illustration of dynamized, active plate in which screws are locked into a sliding element (illustrated in red) within the plate. (**C**) Far-cortical locking screw; the trans-cortex is engaged for construct stabilization while micromotion at the non-threaded screw shaft within the cis-cortex is allowed. (**D**) Variable-fixation locking screw (VFLS); a biodegradable PLGA sleeve positioned below the screw head induces cis-cortical micromotion as it degrades. Image created using Biorender.com (accessed on 1 March 2023).

**Figure 4 bioengineering-11-00525-f004:**
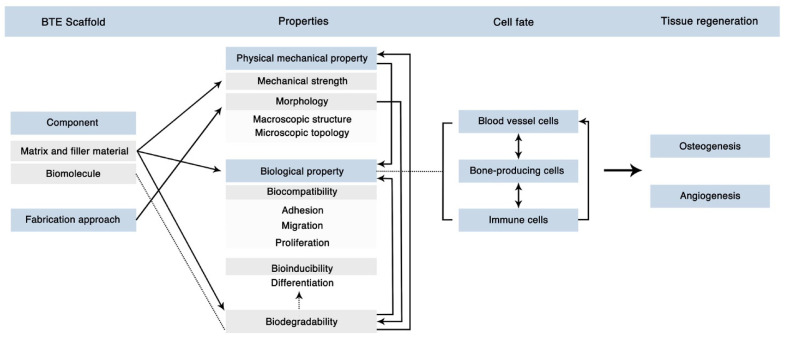
Modern Considerations for Bone Biomaterial Fabrication. Modern bone scaffolds combine biomimetic and biomolecule delivery designs, necessitating both mechanical and biological integrity for proper functionality in vivo. This figure illustrates the complexity and interconnectivity of biomaterial properties to achieve the final goals of bone and vascular regeneration. Figure reprinted from “Figure 12” by Zhu et al. [130], licensed under CC BY 4.0 (https://creativecommons.org/licenses/by/4.0/, accessed on 1 February 2023).

## Data Availability

Not applicable.

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
