# Peer review of "Delayed Union and Nonunion: Current Concepts, Prevention, and Correction: A Review"

_bioengineering, 2024, doi:10.3390/bioengineering11060525_

Round 1

Reviewer 1 Report

Comments and Suggestions for Authors

Manuscript ID: bioengineering-2965252

Type of manuscript: Review

Title: Delayed Union and Nonunion: Current Concepts, Prevention, and Correction: A Review

 Review comments:

This paper describes the biology and biomechanics of delayed bone healing, outlines the known risk factors for nonunion development, and introduces modern preventative and corrective therapies targeting fracture nonunion. While this paper is of clinical interest, authors may consider the issues mentioned below.

1.  Please divide Figure 3 into Figures 3(a) - 3(d), with each illustration showing a fracture gap. Avoid combining all four fixations into a single plate. This will help readers better understand the differences between each type of fixation.

2. The authors may consider adding the concept of bridging plate fixation in Section 2.

Author Response

Thank you for you review comments regarding this manuscript. Attached is our point-by-point response to suggestions. We have uploaded a revised manuscript. 

Reviewer 2 Report

Comments and Suggestions for Authors

 This paper needs major revision based on the below comments.

Also, figure and tables needs to be define.

a literature review table needed

also more comparison with other works.

update reference

What are the current concepts and strategies for preventing delayed union and nonunion in fractures?

Could you provide an overview of the factors that contribute to the development of delayed union and nonunion?

What are the available treatment options for correcting delayed union and nonunion?

Are there any emerging technologies or advancements in the field that show promise for preventing and correcting delayed union and nonunion?

Can you discuss the role of rehabilitation and physical therapy in the management of delayed union and nonunion?

refer to thebelow article

https://www.ncbi.nlm.nih.gov/pmc/articles/PMC8359664/

Comments on the Quality of English Language

 This paper needs majore revision based on the below comments.

Also, figure and tables needs to be define.

a literature review table needed

also more comparison with other works.

update reference

What are the current concepts and strategies for preventing delayed union and nonunion in fractures?

Could you provide an overview of the factors that contribute to the development of delayed union and nonunion?

What are the available treatment options for correcting delayed union and nonunion?

Are there any emerging technologies or advancements in the field that show promise for preventing and correcting delayed union and nonunion?

Can you discuss the role of rehabilitation and physical therapy in the management of delayed union and nonunion?

refer to thebelow article

https://www.ncbi.nlm.nih.gov/pmc/articles/PMC8359664/

Author Response

Thank you for reviewing our manuscript. Attached is our point-by-point response to comments. We have uploaded a revised manuscript.  

Round 2

Reviewer 2 Report

Comments and Suggestions for Authors

Accept